# Evaluating the Chemical Hazards in Wine Production Associated with Climate Change

**DOI:** 10.3390/foods12071526

**Published:** 2023-04-04

**Authors:** Constantin Nechita, Andreea Maria Iordache, Cezara Voica, Diana Costinel, Oana Romina Botoran, Diana Ionela Popescu, Niculina Sonia Șuvar

**Affiliations:** 1National Research and Development Institute for Forestry “Marin Drăcea”—INCDS, 128 Boulvard Eroilor, 077190 Voluntari, Romania; 2National Research and Development Institute of Cryogenics and Isotopic Technologies, ICSI, 4 Uzinei Str., 240050 Râmnicu Vâlcea, Romania; 3National Institute for Research and Development of Isotopic and Molecular Technologies, 67-103 Donat Str., 400293 Cluj-Napoca, Romania; 4Academy of Romanian Scientists, Splaiul Independentei 54, 050044 Bucharest, Romania; 5National Institute for Research and Development in Mine Safety and Protection to Explosion, 32-34 General Vasile Milea Str., 332047 Petroșani, Romania

**Keywords:** ICP-MS, Romanian wines, metallic trace elements, ^87^Sr/^86^Sr and ^206^Pb/^207^Pb isotopic ratios

## Abstract

The climate warming trend challenges the chemical risk associated with wine production worldwide. The present study investigated the possible difference between chemical wine profile during the drought year 2012 compared to the post-drought year 2013. Toxic metals (Cd and Pb), microelements (Mn, Ni, Zn, Al, Ba, and Cu), macroelements (Na, Mg, K, Ca, and P), isotopic ratios (^87^Sr/^86^Sr and ^206^Pb/^207^Pb), stable isotopes (*δ*^18^O, *δ*^13^C, (D/H)_I_, and (D/H)_II_), and climatic data were analyzed. The multivariate technique, correlation analysis, factor analysis, partial least squares–discriminant analysis, and hierarchical cluster analysis were used for data interpretation. The maximum temperature had a maximum difference when comparing data year apart. Indeed, extreme droughts were noted in only the spring and early summer of 2012 and in 2013, which increased the mean value of ground frost days. The microelements, macroelements, and Pb presented extreme effects in 2012, emphasizing more variability in terms of the type of wine. Extremely high Cd values were found in the wine samples analyzed, at up to 10.1 µg/L. The relationship between precipitation and *δ*^18^O from wine was complex, indicating grape formation under the systematic influence of the current year precipitation, and differences between years were noted. *δ*^13^C had disentangled values, with no differentiation between years, and when coupled with the deuterium–hydrogen ratio, it could sustain the hypothesis of possible adulteration. In the current analysis, the ^87^Sr/^86^Sr showed higher values than in other Romanian studies. The temperature had a strong positive correlation with Pb, while the ground frost day frequency correlated with both Pb and Cd toxic elements in the wine. Other significant relationships were disclosed between the chemical properties of wine and climate data. The multivariate statistical analysis indicated that heat stress had significant importance in the chemical profile of the wine, and the ground frost exceeded the influence of water stress, especially in Transylvania.

## 1. Introduction

The 21st century began with severe environmental pollution and global climate change threatening the food chain [1]. The strong relationship between climate variability, wine production, and quality requires novel approaches regarding future perspectives and strategies [2,3]. The literature agrees on the linear relationship between climate warming and heavy plant metal absorption [3]. It sustains the hypothesis of accumulating high metal contents in grapevine. Even if the orogenic heavy metal occurrence is expected in terrestrial environmental matrices, anthropic influence in disrupting the natural equilibrium is very high [4]. Mining activities, industry and agriculture development, and city expansion in the study area released impressive amounts of heavy metals accumulating in soils, aquatic ecosystems, forestry, and agricultural lands [5,6,7,8]. Above all, the impact of contamination on human health represents a key driving factor for food security investigation using various proxy as foodstuffs, spices, or wines [9,10,11].

Wine is a complex matrix containing multiple inorganic and organic components (besides water, sugar, and alcohol) influenced by factors related to the specific production area, such as the grape variety, soil, climate, culture, winemaking, transport, and storage [12,13,14]. The elemental profiling has been used to validate geographic wine origin and to assess multielement fingerprints [13,15,16]. Overall, the total metal content in grapevine originates from natural sources, with a parental rock enriched with metal(oids), and the anthropogenic influences contribute to the site’s discrimination [14,17]. Metal(oid) composition was used to investigate site influence on wine performance [14] or moreover, as a pathway for authenticating wine products [18,19]. The wine elemental composition can influence aging [20] and varies with each step of the vinification process [21]. Most metal(oid) content gradually decreases with processing, except for instances of Cr, Cu, Fe, and Zn, whose content may vary depending on their origin (exogenous or endogenous) [22]. Even the macroelements are strongly correlated, as is the case of high concentrations of K/Mg cations, which typically absorb lower concentrations of Ca [23]. Cation competition results from the plant’s limited absorption of soil elements due to natural availability and fertilization [24]. Human-induced activities such as agricultural practices are the primary sources of Cd, Co, Cu, Mn, Pb, and Zn content in wine composition [25]. The phosphate and nitrogen fertilizer applications are associated with trace elements, such as B, Ba, Cd, Co, Cr, Cu, Mo, Ni, Pb, Rb, Sn, Sr, and V [23]. Other metals, such as As, B, Ba, Cd, Co, Cr, Mo, Ni, Pb, Sb, Sn, Ti, V, and Zn, can originate from biosolids used by farmers, according to different site-specific forms of management at the vineyards [23,26].

During production, the post-harvest sources, e.g., filtration, clarification, and storage, are associated with wine enrichment with Al, Cd, Co, Cr, Cu, Fe, Mn, Ni, Pb, Sr, Ti, V, Zn, and lanthanide elements [14]. Wood aging (post-winemaking treatment) or contact with oak vessels and materials alters the mineral composition and the ^87^Sr/^86^Sr isotopic ratio [27]. Moreover, bentonite treatments are responsible for various elements deposited in the wines [25,28,29]. The geographical traceability of wines was investigated using different methodologies on the basis of multivariate analysis of natural chemical composition data (inorganic or organic parameters) and the ^87^Sr/^86^Sr isotopic signature [30,31,32]. Tracing sources of heavy metal pollution from wines is widely used as a pollution tracer, especially in the case of lead, which is recognized as a severe threat to human and terrestrial environmental health. The isotopic composition of Pb in soils reflects the mixing of all possible sources [33]. The joining of the Pb/Sr isotopic ratio with the deuterium–hydrogen ratio of the methyl, methylene, and stable C and O isotopes responds to questions related to the origins and authenticity of wines from the market [34]. The process of inductively coupled plasma mass spectrometry (ICP-MS) has previously been used for analyzing young-finished, red–white, table-fortified wines and even sweet wines made of over-ripe grape samples in a (semi)quantitative analyte procedure, providing high selectivity, high sensitivity, and lower detection limits in comparison with other multielement techniques [35].

The direct and indirect risks in wine production associated with climate change vary, differing between locations. Above all, the increasing chemical hazard is difficult to control and requires management strategies to conserve product quality and safety [17,36,37]. The present study evaluated the elemental composition of toxic metals (Cd and Pb), microelements (Mn, Ni, Zn, Al, Ba, and Cu), and macroelements (Na, Mg, K, Ca, and P) of twenty-six wines produced in central (Transylvania) and south Romania (Muntenia). The hypotheses tested were (i) whether the wines with provenance from Transylvania and Muntenia are significantly different regarding the metal content and if they are accessible fraudulent products on the national market and (ii) if extreme climate from the year of capitalization 2012 had a significant influence on the chemical parameters of wine that can induce discrepancy in wine discrimination. The isotopic ratios of ^87^Sr/^86^Sr and ^206^Pb/^207^Pb were used as a reference to evaluate possible significant differences between sites. The goal was to assess the variability of metal absorption in all wines on the basis of its location and year of production, with specific attention to extreme climate variables. Another objective was the integration of all data in a multivariate statistical analysis in order to observe possible interrelationships between chemical variables and climate data.

## 2. Materials and Methods

### 2.1. Materials and Reagents

Twenty-six wines available on the market with capitalization years in 2012 and 2013 were investigated. Six vineyards with products highly commercialized were chosen: Corcova (2012), Dragasani (2013, 2012), Jidvei (2012, 2013), Samburesti (2012), Sebes (2012, 2013), and Stefanesti (2012, 2013). The vineyards are geographically located in two distinct regions, Transilvania (Jidvei and Sebes) and Muntenia (Corcova, Dragasani, Samburesti, and Stefanesti). The Carpathian Mountains represent a solid natural barrier in terms of air mass circulation, which is why distinct climatic patterns are noted between the two regions investigated. Five bottles with similar batches were combined to obtain a representative sample that was further analyzed in 2022. Before preparation, the wine samples were briefly stored in a well-ventilated room with a controlled and constant temperature of approximately 14 °C. After the bottles were opened, the samples were immediately distributed to laboratories for multi-element and multi-isotope determinations.

A closed iPrep vessel speed iwaveJ system, MARS6 CEM One Touch, was used to decompose the organic compounds from wines and extract the target elements, according to the two-step temperature-controlled digestion program Microwave Digestion of Wine (CEM Mars 6 Method Note Compendium, 2019). An aliquot (1 mL) of each sample was weighed, followed by digestion in a mixture of wine sample (1 mL) and nitric acid 69% analytical purity (10 mL). Next, the vessel was tightly closed and placed on the rotor, and the microwave temperature was increased up to 210 °C for 20 min and maintained at this level for 15 min. The vessels were cooled and carefully opened at the end of the digestion process. Each digestion was transferred quantitatively with ultra-pure water to a 50 mL volumetric flask.

Ultrapure water with a maximum resistivity of 18.2 MΩcm^−1^, obtained from a Milli-Q Millipore system (Bedford, MA, USA), was used for sample treatment and dilution for ICP-Q-MS analysis. All plasticware was cleaned by soaking for 24 h in 10% HNO_3_ and then with ultrapure water, followed by drying and then ventilation in an oven at 40 °C. High-purity ICP Calibration Standard XXI CertiPUR obtained from CPA chem, Bulgaria, was used for the calibration curve in the quantitative analysis of Mn, Ni, Zn, Al, Ba, Cu, Pb, and Cd. The ICP Multi-Element Standard solution IV of metals Na, Mg, K, Ca, and P 1000 ± 0.002 mg/L (Merck, Darmstadt, Germany) was used for macro elements. The standard solutions were prepared from the stocks by dilution with HNO_3_ solution 0.5% (*v*/*v*). The calibration curve process within 2.5–50.0 μg/L consisted of a blank and five equidistant standards. A correlation coefficient of 0.9999 was a requirement when assessing the method performance.

The instrument performance was optimized using the Tune Solution from Analytik Jena (Jena, Germany). The ^87^Sr/^86^Sr isotope ratio of the resulting extracts was determined by quadrupole inductively coupled plasma mass spectrometry (Q-ICP-MS), after separating strontium and rubidium using cation-exchange chromatography with Dowex 50W-X8 resin and the complexation ability of the carboxylic acid EDTA. An Sr isotope (25 μg/L) standard reference solution (NIST SRM 987) was used as the isotope standard. NIST SRM 987 and SrCO_3_, are certified for their Sr isotopic composition with an ^87^Sr/^86^Sr certified value of 0.71034 ± 0.00026 (2σ) for the NIST SRM 987 Sr strontium carbonate isotopic standard reference material and were used for developing the procedure and evaluating the precision of the obtained Sr isotopic values. Aliquots of extraction solutions were submitted to the separation of Sr and Rb by a cation exchange chromatographic procedure using Dowex 50W-X8 (50–100 mesh) resin (Sigma-Aldrich, St Louis, MO, USA). The chelation of divalent cations was performed using disodium EDTA (Merck, Germany). The Ca and Mg were removed from the matrix by elution with EDTA solution with different concentrations (0.02, 0.05, and 0.1 M) due to the ability of EDTA to complex some divalent cations more strongly and form the cation–EDTA complexes.

A 0.1 g hot dissolved Pb rod isotope standard reference solution (NIST SRM 981) was used as the isotope standard. The NIST SRM 981, Common Lead Isotopic Material, certified for its Pb isotopic composition with a ^207^Pb/^206^Pb certified value of 0.91464 ± 0.00033, was used for developing the procedure as well as for the evaluation of the precision of the obtained Pb isotopic values. The accuracy and precision of the isotope ratio methods for Sr and Pb were tested using replicate ICP-MS Plasma Quant Elite measurements.

### 2.2. Instruments for the Quantitative Determination of Metal Content

Quantitative determination of the HM category of Mn, Ni, Zn, Al, Ba, Cu, Cd, and Pb and macronutrients Na, Mg, K, Ca, and P in wines was performed using a Q-inductively coupled plasma mass spectrometer, ICP-MS Plasma Quant Elite (Analytik Jena, Jena, Germany), equipped with an AIM 3300 Autosampler (Analytik Jena, Germany) and collision-reaction interface iCRI working in H_2_ and He modes.

The optimal conditions were as follows for quantitative multielement analysis: RF power of 1.20 kW; plasma gas flow rate of 12 L/min; auxiliary gas flow rate of 1.50 L/min; nebulizer gas flow rate of 1.05 L/min; H_2_ gas flow rate of 90 mL/min; He gas flow of 120 mL/min; Ar gas flow of 10 mL/min; a dwell time of 500 ms for ^55^Mn, ^60^Ni, ^66^Zn, ^27^Al, ^137^Ba, ^65^Cu, ^114^Cd, and ^208^Pb; macronutrients ^23^Na, ^24^Mg, ^39^K, ^44^Ca, and ^31^P; and ^209^Bi, ^6^Li, ^45^Sc, ^159^Tb, and ^89^Y isotopes that served as internal standards for quantitative analysis mode. The standards were linear over the concentration range of 2.5 to 50 μg/L, as assessed by linear regression with correlation coefficients between 0.9991 and 0.9999. The limits of detection, depending on the element, ranged between 0.1 and 0.6 μg/L for the microelement determinations and 0.05 and 5.0 mg/L for the macroelements. The global recovery for each element (R) was estimated, and the obtained values were between 85 and 105%. The accuracy of the methods was evaluated by replicate analyses of fortified samples (5 μg/L and 10 mg/L concentrations), and the obtained values ranged between 0.9 and 15%, depending on the element.

Strontium and lead isotope ratios in wine were accomplished with the same quadrupole inductively coupled plasma mass spectrometry (Q-ICP-MS) spectrometer (ICP-MS Plasma Quant MS Elite, Analytik Jena, Jena, Germany). The elemental profile was coupled with specific element ratios (K/Rb, Ca/Sr) and an ^87^Sr/^86^Sr isotopic ratio, which evaluated the possibility of discriminating the wines according to their geographical origin. The optimal conditions were as follows in quantitative analysis mode: RF power of 1.28 kW, plasma gas flow rate of 9 L/min, nebulizer gas flow rate of 1.01 L/min, a dwell time of 500 ms, an integration time of 169 s, and scan mode peak hopping. The analytical values obtained for NIST SRM 987 for the ^87^Sr/^86^Sr isotopic ratio ranged from 0.7100 to 0.7117, with a mean isotope ratio of 0.7107, RSD (%), with a mean of 0.0648%. The average accuracy was 0.0131%, and the precision was 0.0215%. The analytical values obtained for the common Lead Isotopic Standard NIST SRM 981 for the ^207^Pb/^206^Pb isotopic ratio ranged from 0.91431 to 0.91496, with a mean isotope ratio of 0.91453, and the precision was 0.0404%. Thus, performing data acquisition using 5 points and 10 scans per peak of 10 replicates resulted in satisfactory resolution measurements.

### 2.3. Isotopic Analysis

The ^18^O/^16^O isotopic ratio of both the water extracted from wine and ^13^C/^12^C, (D/H)_I_ and (D/H)_II_ isotopic ratios in the ethanol extracted from wine were examined by following the methods established by the International Organization of Vine and Wine (OIV), MA-AS2-12, MA-AS312-06, and MA-AS-311-05. All the wine samples were distilled using an automated Cadiot distillation column unit (ADCS Eurofins, Nantes, France) to obtain ethanol with an alcoholic strength of at least 92% w/w and a yield of 90%. δ^18^O was measured using a continuous-flow isotope ratio mass spectrometer (CF-IRMS) Delta V Plus (Thermo, Bremen, Germany) coupled with a GasBench II isotopic equilibration module, and the results were processed using Isodat 3.0 software. An aliquot (500 µL) of water extracted from wine was equilibrated for 20 h at 24 °C with a gas mixture (0.36% CO_2_ in helium). The samples were analyzed comparatively with laboratory standards IA-R052, IAR053, and IA-R054 provided by the ISO Analytical Laboratory Standard, UK.

For quality control, certified reference materials, such as BCR 656 and BCR 660 provided by the Institute for Reference Material and Measurements (IRMM), Belgium, were analyzed together with the wine samples. For δ^13^C analysis, 0.1 µL of ethanol was injected into and combusted in a Flash EA1112 HT elemental analyzer (ThermoFisher, Bremen, Germany). Then, the CO_2_ gas was flushed via a continuous helium flow into a Delta-V IRMS (ThermoFisher, Bremen, Germany) and processed for their stable isotope ratio using Isodat 2.5 software. For both isotopic measurements of δ^18^O and δ^13^C, the enlarged reproducibility of measurements of the whole experimental procedure was ±0.3‰. The results were expressed in the δ-notation and ‰ as the Vienna-Standard Mean Ocean Water (VSMOW) and the Vienna-PeeDee Belemnite (VPDB) standards for oxygen and carbon, respectively.

The measurement of deuterium–hydrogen (D/H) ratios on methyl CH_2_DCH_2_OH(I)–(D/H)_I_ and methylene CH_3_CHDOH(II)–(D/H)_II_ positions in wine ethanol was performed using the SNIF-NMR method. The SNIF-NMR spectra were recorded on an Ascend 400 Bruker spectrometer equipped with a 10.0 mm selective deuterium probe-head tuned to a frequency of 61.42 MHz, with a fluorine lock and an automatic sample changer. Each sample was measured using 200 scans. The measurements were performed according to the OIV protocol using N,N-Tetramethylurea (TMU) from the Institute for Reference Materials and Measurements at Geel, Belgium, with a known isotopic composition employed as an internal standard. Hexafluorobenzene (Aldrich) was further added for the lock signal (19F). The results, mean values, and standard deviations were calculated with Eurospec (Eurofins-Nantes) software from 10 repetitive experiments with an exponential multiplying factor (LB) equal to 2. The isotope ratios of a given position are expressed in parts per million on the international scale VSMOW. The average precision values of the ratio measurements were ±0.6 ppm for (D/H)_I_ and ±0.8 ppm for (D/H)_II_. All isotopic measurement methods are accredited according to EN ISO/IEC 17025:2018, confirming the laboratory’s ability to perform valid and comparable stable isotope results.

### 2.4. Climate Dataset

We analyzed the temperature (maximum temperature (tx °C), minimum temperature (tn °C), and mean temperature (tg °C)) and precipitation (rr mm/day) to evaluate the primary climate conditions. The mean monthly data were extracted from two climate open-access databases: CRU TS 4.05 and E-OBS v23.1e. Ground frost day frequency (frs days), the self-calibrating Palmer Drought Severity Index (scPDSI), and the standardized precipitation–evapotranspiration index (SPEI z-values) explain the extreme local events. The number of days when the nighttime temperature decreased below freezing represents the frost days and indicates multiple ecosystem changes. The IPCC Third Assessment Report showed that the number of frost days would reduce under climate change, raising several negative impacts on the grapevine [38]. Since sustained drought affects grape quality [39], we used two indices to evaluate the drought (scPDSI and SPEI). Firstly, the scPDSI is a standardized index that spans between −10 (dry) to +10 (wet) and has a long-term memory of previous climate conditions (and thus it is less sensitive to drought recovery) [40]. Secondly, the SPEI is very sensitive to the starting and evolution of drought events, with values less than −2 indicating extreme events [41]. The SPEI can be evaluated for different interval scales, and were chosen here to investigate the cumulate defect of 1, 3, 4, and 6 months.

### 2.5. Statistical Analysis

The one-way ANOVA was performed to determine the mean comparison and equal variance between isotopic and element distribution at the *p* < 0.05 probability level with Tukey’s and Levene’s tests. The factor analysis was performed using the principal component method with varimax rotation to analyze the correlation matrix, for which three factors were extracted. Partial least square analyses were performed using Wold’s iteration with variables scaled, for which a maximum of 15 factors were extracted. The relationships between climate variables were tested to evaluate the degree of the correlation between factors.

## 3. Results and Discussion

### 3.1. Climate Characterization

Climate warming poses a significant risk associated with the increasing frequency, severity, and duration of extreme events and actions currently described as biodiversity loss, degradation, and transformation of natural ecosystems worldwide. The year 2012 was one of the warmest years experienced, being 0.6 °C warmer than the mid-twentieth-century baseline. During the drought year 2012, the mean annual temperatures in Sebes and Corcova sites were most contrasting (9.12 ± 9.91 °C and 12.15 ± 10.52 °C, respectively), with extremely positive values in July (22.68 °C and 26.27 °C, respectively). The post-drought year 2013 showed a cold July and the hottest August out of all sites, and the spring monthly mean temperature was higher than in 2012. On the basis of the annual maximum temperature, we noted a variability between sites from 27.04 °C (Sebes, 2013) to 34.59 °C (Dragasani, 2012), which indicated a more arid climate in Muntenia compared with Transylvania. The largest difference between the 2012 and 2013 climate datasets was found to be the maximum temperature (Figure 1A). Interactions between the warmer weather in 2012 and the sector-based approach suggested a new dimension in disaster risk, exposure, and vulnerability [42]. Thus, it was demonstrated that increasing extreme temperatures impact fruit composition and wine development due to the resulting higher pH and sugar content and the lower acidity [43]. Even the mycotoxins, contaminants, and residues correlated strongly with environmental conditions [44,45] and were directly involved in grapevine proprieties. The implications of drought events are associated with differences in the accumulation of microelements, macroelements, and toxic elements in wine [46].

It was observed that the precipitation deficiency had a slight shift in occurrence associated with translating the maximum amount from spring to autumn, as was also discussed in the region using different proxies [47]. The highest site differences between precipitation in 2012 and 2013 were at Dragasani (161.1 mm/year), followed by Sebes, Jidvei, Corcova, Samburesti, and Stefanesti (141.3, 116.9, 102.2, 88.9, and 51.1 mm/year, respectively). Therefore, the climatic conditions in 2013 were characterized by moderate drought. During spring and early summer (March, May, and June), the extreme drought evaluated through SPEI1 occurred at all sites only in the year 2012. The SPEI3 indicated high drought climate conditions in August–September compared with the SPEI4, for which the trend was shifted by one month from September to October. We noted a significant difference between mean climate conditions in Jidvei in 2012–2013 when analyzing the SPEI4 and SPEI6. The highest scPDSI values were calculated in 2012 for Jidvei between July and December (−4.05, −4.92, −5.15, −4.88, −5.06, and −4.47), followed by Corcova and Sebes from September to December and Dragasani and Samburesti during September and November. The literature discusses the role of the temperature in October–November due to it being the period with the most intensive growth of absorbent roots and the highest amount of accessible groundwater that is essential for plant growth and development [48]. The temperature forcing in June shifts fruit ripening from the hot summer to cold autumn (October–November), associated with smaller fruits, lower pH, higher acidity, and a disrupted content of total phenolics, flavonoids, and anthocyanins [49,50]. According to scPDSI indices, the early growing season was most affected in 2012 by drought, with possible chances to induce changes in grapevine proprieties (Figure 1B). In 2013, scPDSI values indicated an extreme drought climate for Sebes during January–February (−4.49; −4.34) and at Jidvei in January (−4.14). Drought events from early spring to autumn were present in both years but with different intensities and occurrence periods, illustrating the legacy effects of the demand for freshwater resources. A significant difference in the mean and variance of Jidvei versus Stefanesti (2012) and Jidvei versus Sebes (2013) showed a high variability between sites, even at the same year of production. Climate warming influences wine manufacturing due to its high impact on fruit composition and vine formation under changing water status, solar radiation, and thermal increase. High sugar content modifies the phenolic maturity, color density, and poor aroma, reducing the organoleptic and quality of the final product [51].

Another challenge for future grape wine distribution and its impact on nutrient composition is represented by post-budburst freeze damage. February (Corcova and Dragasani) and March (Samburesti, Stefanesti, Jidvei, and Sebes) was the last spring period with a negative minimum temperature. The ground frost day frequency indicated a decreasing tendency in 2013 compared with 2012 in all sites, even if the mean value of frost day count increased (Figure 1C). The effect of factors cumulated in frost damage levels and include fluctuation in the temperature (which was higher in 2012), the force of the wind, the age of plants, and the physiological state [52]. On the basis of the results identified in the literature, the most resistant varieties to cold climate are Riesling and Pinot Noir [53]. Between sites, we recorded a higher rate of frost day frequency occurrence at Jidvei and Sebes than at the other vineyards. Even the earlier budburst caused by climate change conditions is not negligible, potentially delaying the activity of pollinating insects and increasing the susceptibility to an early frost [54]. Worldwide literature has concluded that abnormal dieback in *Vitis vinifera* L. in the last decade was caused primarily by the high sensitivity to minor changes in water reduction and increasing evapotranspiration, which alter even the macro and microelemental profile [55]. Drought will significantly affect south-eastern Europe in the coming years, visible mainly in terms of heatwave frequency and intensity [56]. Unfortunately, the implications of climate change in the wine sector are adjusted in Romania through one-dimensional legislation and often by a lack of practical measures, which highlights future implications for vineyards considering the opportunity of cultivating higher regions due to changing climate conditions [57].

### 3.2. Multielement Wine Content

Mineral elements mediate physiological and biochemical processes, including protein synthesis, photosynthesis, enzyme activation, and osmoregulation. Drought stress can be avoided by the proper content of mineral elements, which participate in several defense mechanisms such as antioxidant and osmotic regulation [58]. All microelements (Mn, Ni, Zn, Al, Ba, and Cu) had a maximum content in 2012, emphasizing more variability based on the type of wine (Figure 2A). Thus, the maximum values decreased in the following order Mn > Al > Cu > Zn > Ba > Ni, considering Cabernet Sauvignon (Mn = 1843 µg/L, Al = 1641 µg/L, Ba = 465 µg/L), Riesling (Ni = 102 µg/L), Merlot (Zn = 580 µg/L), and Muscat Ottonel (Cu = 860 µg/L). Our results show comparable values with similar studies conducted in Romania, indicating minor changes in elemental profiles concerning the region and yearly variability [15,16,59,60]. Moreover, we noted that Romanian wine contains high values of macronutrients compared with worldwide reports [17]. The comparison between 2012 and 2013 illustrated high extreme values in drought years only for K, as well as values that were almost similar for Na, Mg, Ca, and P (Figure 2B). The macroelements had a decreased order from K > P > Mg > Ca > Na (437, 105, 84, 50, and 18 mg/L, respectively), and the maximum values were comparable with previous studies from Romania for Mg (133 mg/L), Ca (70 mg/L), K (520 mg/L), and P (142 mg/L). It is worth mentioning that all maximum macroelement contents were found in the assortments of Pinot Noir, with 2012 as the year of capitalization. Even if the high K values were associated with equilibrium in the red color pigment complex with anthocyanin and tartaric acid, the present study did not establish significant statistical differences between the profiles of the wine. Between wine types, we noted low variability of macroelement occurrence, exemplifying the high concentration in Chardonnay (Na = 21 µg/L, P = 104 µg/L), Merlot (Mg = 96 µg/L), Cabernet Sauvignon (K = 470 µg/L), and Pinot Noir (Ca = 60 µg/L), and low concentration in Cabernet Sauvignon (Na = 14 µg/L), Chardonnay (Mg = 70 µg/L), Pinot Noir (K = 420 µg/L), and Merlot (Ca = 46 µg/L, P = 98 µg/L). A differentiation between geographical location and microelements showed the occurrence of extreme values in Jidvei (Mn = 1843 µg/L, Al = 1641 µg/L, Cu = 860 µg/L), Sebes (Ni = 102 µg/L), Samburesti (Zn = 580 µg/L), and Stefanesti (Ba = 465 µg/L). Regarding the macroelements, Na and Mg were measured as having extreme levels in Jidvei (26.75 and 133 µg/L, respectively), and K, Ca, and P were found in Dragașani wine (520, 70.32, and 142 µg/L, respectively). The Cu, Zn, Al, Ba, Mn, Sr, and Pb were mainly associated with soil-derived elements, being fingerprints for chemical traceability in grapes and wine [61]. Other elements, such as Cr and Ni, are frequently associated with stainless steel contact during the tank-fermented Charmant production method [62].

It can be discussed that from the six types of Jidvei wine analyzed, we found high differences between the level of toxic metal concentrations, with the maximum values in Cabernet Sauvignon (Cd = 8.01 µg/L; Pb = 155 µg/L) and the lowest in Chardonnay in the case of Cd (0.09 µg/L) and Traminer rosé for Pb (80 µg/L). On the other hand, we found the highest value of Cd in the Stefanesti region for Riesling wine (10.1 µg/L), being extreme in comparison with other areas, e.g., Germany, Greece, Spain, the Czech Republic, Hungary, Ethiopia, Turkey, Croatia, Italy, and Poland [17,63,64]. Since similar values (in the case of Pb) were reported for Romania [15,16,59], we consider that industrial pollution and road traffic are the main contributors to toxic elements in the investigated wines. Even so, brass components in wine processes or phosphate fertilizers are not excluded from the multiple causes of the enrichment of wine with toxic elements. Generally, the metal concentration in wine contains exogenous influences such as the parental signature, fertilization practices, climate conditions, transboundary and local pollution, or technological processes [21]. However, a comparison between worldwide reports regarding metal concentration showed various patterns, from low amounts [65] to high concentrations [66,67,68]. The present study shows high variability of Pb in 2012 compared with 2013 (Figure 2C).

### 3.3. Isotope Signature of δ^18^O, δ^13^C, D/H, ^87^Sr/^86^Sr, and ^206^Pb/^207^Pb in Wine

The wine origin was tested using *δ*^18^O stable isotopes due to oxygen involvement in plant metabolism and fractionation processes during plant growth. The samples investigated revealed that *δ*^18^O of wine varied from 0.32‰ to 5.99‰ (Transilvania) and −1.09‰ to 7.39‰ (Muntenia). The minimum values were associated with a cooler climate, and the maximum (Jidvei, Merlot, 2012, and Samburesti, Merlot, 2012) indicated a higher variability of *δ*^18^O of wine water between regions. The Tukey test was used to compare the means of the two areas and no significant differences were found. Lower variability was found in the post-drought year 2013 at 0.32–5.60‰ (Transilvania) and 0.46–4.66‰ (Muntenia) (Figure 3A). The Muntenia region was found to have a higher dispersion of *δ*^13^O of wine water values compared with Transylvania, even if the mean was comparable (3.45 ± 2.36‰ and 3.72 ± 1.64‰, respectively). A significant difference between *δ*^18^O in 2012 and 2013 was determined in Jidvei Sauvignon Blanc and was 5.40‰ and 0.32‰, respectively. In contrast, the Sebes Sauvignon Blanc wine presented a more depleted value for *δ*^18^O in 2012 (3.45‰) compared with 2013 (5.60‰). The relationship between precipitation and oxygen isotope from wine was complex, indicating grape formation under the predominant influence of *δ*^18^O values from precipitation during the current year of fruit formation. A significant correlation was found between *δ*^18^O and precipitation during all months (*p* < 0.01). Differentiating the regions, we found that the relationship between *δ*^18^O and precipitation was negative below the limit of significance (*p* < 0.05) for Transylvanian samples and positive and significant for Muntenia wines with a maximum in August (*r* = 0.82; *p* < 0.01). We found differences in 2012 and 2013 (3.95 ± 2.02 and 2.96 ± 2.96‰, respectively) which were also discussed for Romanian wines in 2008 and 2009 (3.5 ± 0.8 and 4.0 ± 1.7‰, respectively) [69].

The (D/H)_I_ ratio is regularly used to evaluate wine adulterations through various methods, such as mixing wines with different qualities or adding sugar before capitalization [70,71]. For a correct assessment, it is necessary to evaluate this parameter coupled with *δ*^13^C to reduce the uncertainty limit. The EU wines range between 98 and 104 ppm [72,73], as was indeed observed even in our study (98.7–105 ppm). There was no pattern to discriminate the years of production on the basis of (D/H)_I_ (Figure 3B). The maximum value for 2012 and 2013 was at Sebes, with an insignificant difference (103.6 and 103.5 ppm) compared with the minimum ratio, which was calculated in 2012 at Jidvei (98.7 ppm) and Stefanesti (99.1 ppm). The (D/H)_II_ values varied on the basis of the isotopic composition of the fermented water, indicating values between 119.8 and 131.5 ppm. For the differentiation between the raw materials and to identify illicit adulteration, even the (D/H)II values can be used as they are usually higher than 125 ppm for fruit and wine ethanol due to the enrichment of deuterium in fruit water [74]. The ethanol from sugar or other vegetables usually has a lower (D/H)_II_ [75], and mixing with water before fermentation induces the decreasing ratio. The differences in C_3_ plant (grapevine) photosynthesis (Calvin photosynthetic cycle) are dependent on water availability, resulting in less *δ*^13^C than compared with C_4_ plants (Hatch–Slack cycle) [76]. Therefore, the stable carbon (mainly *δ*^14^C) needs to be more questionable in evaluating the efficiency of wine adulteration due to nuclear tests since 1960 which increased the incertitude. On the basis of stable carbon (*δ*^13^C of ethanol) isotope ratios of wine samples, we found differences between Transylvania (−27.74 to −24.56‰) and Muntenia (−28.21 to −24.06‰). Furthermore, we observed that *δ*^13^C had disentangled values between years, as was the case for Sauvignon Blanc at Jidvei (−24.6‰ (2012) and −27.74‰ (2013)) compared with Sebes (−26.18‰ (2012) and −24.56‰ (2013)).

The ^87^Sr/^86^Sr signatures are considered geological fingerprints for tracing geographic provenance tracers without consistent information regarding yearly variability [77]. The ^87^Sr/^86^Sr values in the analyzed wines varied between 0.705 (Corcova, Merlot, 2012) and 0.773 (Sebes, Riesling, 2012) (Figure 3C). On the basis of regions, the highest mean values (± standard deviation) were found for Stefanesti (0.768 ± 0.002), followed by Jidvei (0.732 ± 0.009), Samburesti (0.729 ± 0.007), Sebes (0.720 ± 0.006), and Corcova (0.707 ± 0.002). Previous studies indicate similar origins, and possible differences between years are assimilated with contamination or combining grape products from large areas [78]. Analyzing only one site (Jidvei), the Sr in 2012 and 2013 had values almost similar, at 0.732 ± 0.009 and 0.738 ± 0.008, respectively. Comparing our results with those conducted in eastern and south Romanian vineyards for the 2012–2016 harvest, we noted a high ^87^Sr/^86^Sr, compared to previously reported 0.710–0.723 [49], 0.702–0.768 [69], and 0.701–0.742 [70]. The ^206^Pb/^207^Pb isotopic signature was high, varying between 1.161 (Stefanesti, Sauvignon Blanc, 2013) to 1.256 (Samburesti, Cabernet Sauvignon, 2012), indicating an influence of environmental pollution. The purification and filtration agents from enological practice (use of bentonites; a mixture of bentonites, active carbon, and diatomaceous earth; and membrane filtration) contain low Pb concentrations, which in our case should be discussed. The past decades’ changes in European atmosphere and lithosphere conditions have been extensively studied using various wine species and the Pb tracer, indicating changes in the ^206^Pb/^207^Pb isotopic signature [33]. Our results show difficulty separating the geogenic background from atmospheric pollution, and extreme values are associated with non-environmental contamination sources (Figure 3C). The ^206^Pb/^207^Pb in wines are comparable to those measured in France, where the isotope ratio ranged between 1.151 and 1.191 [33].

### 3.4. Multivariate Statistics

Pearson’s correlation matrix demonstrated significant dependencies between climate variables and the chemical parameters measured in wine samples (Figure 4). Generally, the Palmer Drought Severity Index (March, April, June, July, and August) and ground frost day frequency (February) had no significant correlation with toxic metals, micronutrients, macronutrients, and stable isotopes. The scPDSI in February correlated negatively with Mg (*r* = 0.69) and positively with (D/H)_II_ (*r* = −0.69). The deuterium–hydrogen ratio was previously associated with climate variables (temperature and humidity) for wines analyzed in Swiss vineyards (Valais and Graubünden) in the year 2000 [79]. Late frost from May and April had a significant positive relationship with the ^87^Sr/^86^Sr ratio, Na, and (D/H)_I_ and a negative association with the ^206^Pb/^207^Pb ratio, Al, Mn, Cd, and Pb. A robust negative association (*p* < 0.001) was revealed between latitude versus the ^206^Pb/^207^Pb isotopic ratio (*r* = −0.62) and was positive for longitude versus (D/H)_I_ (*r* = 0.81).

The correlation matrix described the significant interrelationship between parameters, indicating a similar origin and accumulation patterns in grapevine. Metal concentrations in grapevine rely on metal speciation, soil parameters, and environmental parameters characteristic for each plant type. In contrast, plant growth and phenology control micro and macronutrient uptake in plants [80]. This study presented specific interrelationships between elements, indicating the possible positive relationship between toxic metals (Pb and Cd), Mn (*r* = 0.66; *r* = 0.60), and Al (*r* = 0.55), and a negative relationship with Na (*r*= −0.40). The common source of these metals corresponds to the application of fungicides, pesticides, and fertilizers. Their intensive application for the long term is responsible for adverse environmental effects, transferable to the soil, water, and organisms [10,81]. The increased concentrations of Al in wine are associated with bentonite use for fining [82], but also with herbicide and insecticide use [83] or wine production and storage [19]. A strong correlation between Cd, Pb, and Al was found in wine produced in vineyards near road traffic, especially with industrial destinations [82]. We observed that Na was positively (*p* < 0.01) associated with the deuterium–hydrogen ratio of the methyl ratio (*r* = 0.57) and negatively with Mn (*r* = −0.49) and Ni (*r* = −0.45). Sodium has anthropogenic sources from salting treatment and fertilizer and has a natural origin in sea salts [84]. The *δ*^18^O correlated positively with Mg and negatively with the ^87^Sr/^86^Sr isotopic ratio. This positive relationship indicates the ability of Mg to influence the discrimination and classification of wine origins on the basis of maximum content [85]. A specific relationship was tested between the (D/H)_I_ ratio and several elements associated with the investigation of wine origin and traceability, namely, Cd (*r* = −0.41), Mn (*r* = −0.50), and ^206^Pb/^207^Pb (*r* = −0.40). However, Mn correlated significantly with Al (*r* = 0.60) and Mg (*r* = 0.43) and negatively with Na (*r* = −0.49), and high values were associated with bentonite use. Cu and Zn correlated positively with Mg (*r* = 0.45 and *r* = 0.40, respectively), indicating both natural and anthropogenic origins of macro and microelements. The results obtained in the current study are comparable with similar values discussed for Romanian wines [16,59,60,86].

PLS-DA analysis was used to display the natural grouping of elements and wine origins [87], even if the wines needed to be better distinguished between the 2012 and 2013 harvest years (Figure 5). The Transylvanian wine is well-differentiated from Muntenia on the basis of appropriate geological and climate conditions. The PLS-DA analysis transformed the class vector into a dummy matrix Y, which contains the membership of each sample in binary form. Furthermore, the PLS2 model was calibrated on the basis of the Y matrix, and the probability of the sample assigning to one class was calculated on the basis of the estimated class values [88]. Each modeled class corresponds to a classification function with linear combination coefficients of the original variables to define the classification score. The variance explained for X effects was 47.74% (factor 1), 34.10% (factor 2), and 8.96% (factor 3), resulting in more than 90% cumulative X variance. The variance explained for Y responses was 12.69% (factor 1), and cumulative variance for the first 10 factors did not overhead 55.52%. The discriminating role of each variable was obtained by analyzing their importance at the classification coefficients of the PLS-DA models. Only the maximum temperature, minimum temperature, self-calibrating Palmer Drought Severity Index in February, and ground frost day frequencies in March and May exceeded the threshold of significance. The results indicated that not all environmental factors could induce significant associations between class values. Even so, the most considerable parameter except extreme temperature was the ground frost day frequency in May. The frost duration and strength of frost were significantly affected by the tolerance of grape varieties, and we noted an increasing mean number of days with frost in 2013 compared with 2012. The Transylvanian vineyards are subjected to an increased risk of spring frost. Similar results were reported for other temperate regions, as in the cases of Hungary [36], Serbia [89], southeast England [54], France [90], China [37], and Canada [2]. PLS-DA analysis suggests a possible reduction in accuracy class predictions based on environmental factors due to natural adaptations more evident in the Pinot Noir assortment and human-induced practices (delayed winter pruning).

Factor analysis was carried out for all data analyzed in the experiment to characterize wine from Transylvania and Muntenia (Figure 6A). The wine microelements, macroelements, toxic elements, climate data, stable isotopes, isotopic ratio, and deuterium–hydrogen ratio showed that the first three dimensions described only 52.96% of the total variance (F1 = 29.75%, F2 = 13.30%, and F3 = 9.90%). The variables that defined F1 were scPSDI_May, frst_Mar, frst_Apr, frst_May, (D/H)_I_, Cd, Pb, Mn, Al, Na, ^87^Sr/^86^Sr and ^206^Pb/^207^Pb ratios, and tx. The Cd, Pb, Mn, and Al values exhibited opposite extreme values, indicating their decreasing dependence. On the other hand, Cd, Pb, the ^206^Pb/^207^Pb ratio, and tx were located in the negative F1/F2 plane. This group associated the major climatic factors with toxic elements, microelements, macroelements, and proprieties associated with wine traceability, which were all highly correlated. Even so, considering all the interrelations discussed, it is worth mentioning that the temperature strongly correlated with ^87^Sr/^86^Sr and ^206^Pb/^207^Pb isotopic ratios and Pb and Mn elements. This behavior must be considered in terms of the current extreme climate occurrence since it may be essential in the discrimination of wine under expected increasing variability, mainly due to plant adaptation to future environmental conditions [91,92]. Large-scale changes in the Pb and Sr isotopic ratio in early Earth stages were associated with temperature corresponding to an *δ*^18^O of around −39‰ [93]. Nowadays, more complex conditions must be considered due to fast environmental changes associated with anthropogenic environmental conditions that can induce discrepancies in the correlation between wines and labile fractions. Various influences are responsible for enlarging the discrepancies in ^87^Sr/^86^Sr and ^206^Pb/^207^Pb isotopic ratios in soil, grape, pulp juice, skin, and seeds during different stages of growth and development [31]. F2 grouped *δ*^13^C, *δ*^18^O, (D/H)_II_, Cu, Mg, Ca, and P. Except for the deuterium–hydrogen ratio of the methylene versus Mg, all parameters had a significant positive correlation. The results confirm a strong relationship between macroelements and stable isotopes and their role in the discrimination of wine produced in different regions [94,95]. The variables with the most significant weight in factor three contained Ni with positive values in the F2–F3 plane and was negative in the F1 plane, and Zn was positive in all three planes.

Cluster analysis was used to identify similarities between wine parameters and climatic factors using those variables that significantly correlated as the input data (Figure 6B). The results show that the first cluster was associated with ^206^Pb/^207^Pb, Pb, Cd, Mn, and Al together with tx, tg, and tn. The most representative variable was the maximum temperature and the least representative variable was aluminum. The second cluster was based on the most representative variables Mg, Cu, Ca, P, Zn, Ba, *δ*^13^C, and *δ*^18^O. The second cluster associated variables on the basis of their correlation with February values of the self-calibrating Palmer Drought Severity Index. From the same branch dissociated around the macroelement K was the Ni and deuterium–hydrogen ratio of the methylene. The fourth cluster was formed on the basis of ground frost day frequency in May and included Na and the deuterium–hydrogen ratio of the methyl. The atmospheric precipitation represents a dominant force on the ^87^Sr/^86^Sr isotope ratio, which can also be correlated with a higher isotopic ratio obtained in the present study in comparison with others conducted in the same regions. The cluster analysis helps to understand which microclimate in a significant way controls the micronutrients, macronutrients, isotopic ratio, and stable isotopes in commercial wine produced in Transylvania and Muntenia.

## 4. Conclusions

Various environmental factors strongly influence wine production, but climate change is responsible for various alterations, including increasing chemical hazards. The present survey investigated possible changes in the chemical composition of wine between the drought year (2012) and post-drought (2013). According to scPDSI indices, the Muntenia sites were most affected by drought in the early growing season of 2012, and the mean ground frost day count increased in 2013 in Transylvania. Minor changes in elemental profiles were found between years, and only K was found in an extremely high amount in 2012. Here, we report an extreme value of Cd (Stefanesti region) in Riesling wine, also noting that the Pb had a concentration above the values mentioned in the literature (‰). A significant difference between *δ*^18^O in 2012 and 2013 was determined in Jidvei Sauvignon Blanc. On the basis of stable carbon (*δ*^13^C of ethanol and glycerol) isotope ratios of wine samples, we found differences between Transylvania and Muntenia and also noted disentangled *δ*^13^C values between years. The influence of environmental pollution on sites investigated was associated with yearly changes in ^87^Sr/^86^Sr and ^206^Pb/^207^Pb isotopic patterns. A strong significant correlation was found between climate variables and chemical profiles. Nevertheless, wine is a low-risk product, with the local enrichment of various chemicals, and no significant differentiation in terms of the elemental profile was achieved between years with clearly different climatic patterns.

## Figures and Tables

**Figure 1 foods-12-01526-f001:**
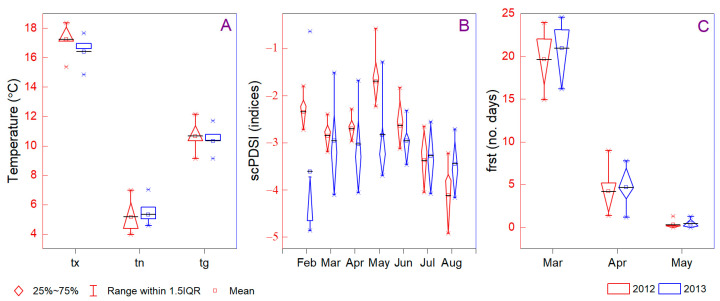
Mean climate differences between 2012 and 2013, according to the mean yearly record of each data grid; (**A**) temperatures; (**B**) self-calibrating Palmer Drought Severity Index; (**C**) ground frost day frequency. The 2012 (red) and 2013 (blue) values are illustrated.

**Figure 2 foods-12-01526-f002:**
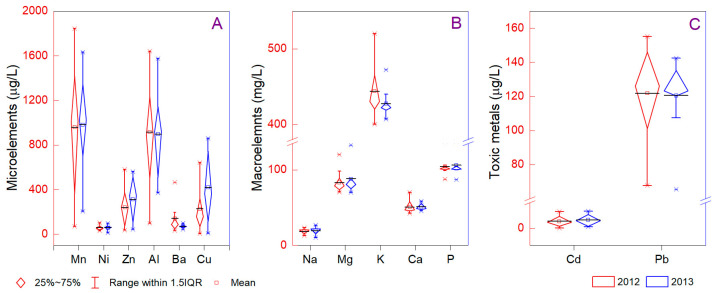
The distribution of microelements (**A**), macroelements (**B**), and toxic elements (**C**) of 2012 and 2013 wine samples. Comparatively, the values from 2012 (red) and 2013 (blue) are illustrated.

**Figure 3 foods-12-01526-f003:**
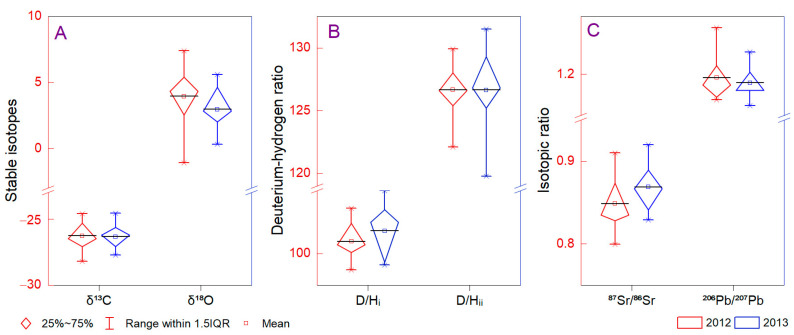
The distribution of stable isotopes (**A**), the deuterium–hydrogen ratio (**B**), and the isotopic ratio (**C**) of wine capitalized in 2012 and 2013 in Transylvania and Muntenia. Comparatively, the values from 2012 (red) and 2013 (blue) are illustrated.

**Figure 4 foods-12-01526-f004:**
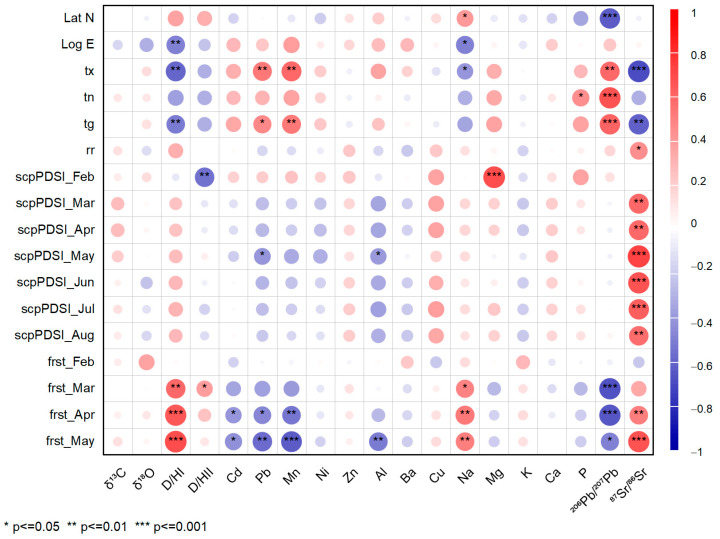
Pearson’s correlation coefficient matrix between the climate, geographical position, the year of capitalization, micro and macroelements, toxic elements, and isotopic ratios. The correlation intensity was illustrated from insignificant correlation coefficients (smaller circles and transparent colors) to significant ones (higher diameters and dark colors). The confidence interval of the significant values was presented differentiated between 95%, 99%, and 99.9%.

**Figure 5 foods-12-01526-f005:**
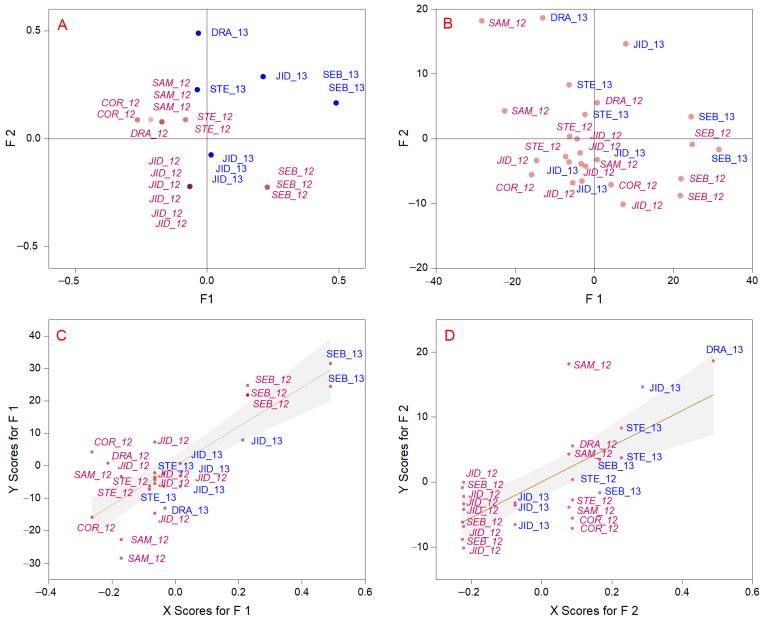
Discriminant analysis (PLS-DA) score plot of the first two principal components. The 95% confidence ellipse is included in the score plot. (**A**) X score plot; (**B**) Y score plot; (**C**) fitted X1-Y1 score plot; (**D**) fitted X2-Y2 score plot. The names represent the abbreviation of the vineyard (the first three letters) followed by the capitalization year, marked with read (2012 year) and blue (2013 year).

**Figure 6 foods-12-01526-f006:**
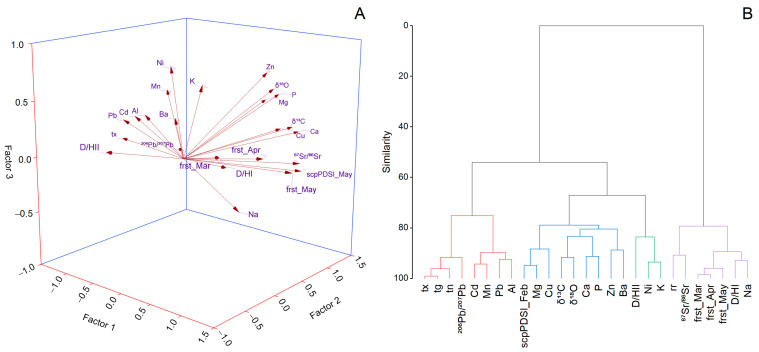
Graphical representation of the first three factors extracted from the factor analysis on the basis of the wine parameters analyzed and environmental variables (**A**). The hierarchical cluster dendrogram resulted from the correlation matrix of the environmental variables and microelements, macroelements, toxic element stable isotopes, and isotope ratio for the Romanian wine analyzed (**B**).

## Data Availability

All data relevant to the study are included in the article.

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
