# Peer review of "Evaluating the Chemical Hazards in Wine Production Associated with Climate Change"

_foods, 2023, doi:10.3390/foods12071526_

Round 1

Reviewer 1 Report

This an interesting manuscript but the authors have over cited literature thus shadowing the relevance of the manuscript. The authors should go through the manuscript and remove many trivial sentences and revised the language. The figures should be improved. Below are some suggestions to help improved the manuscript.     

Line 17-18 the objective is not clear. Please rephrase to improve clarity

Line 19  A database that cumulated….meaning what? The abstract needs to be looked at. The current form is difficult to follow couple with grammar issues

The introduction is very long please cut some out. 25 citations in introduction alone is weird

Line 114 to 123 please move elsewhere. Not appropriate after stating the aim of the study

Line 127 assuming? Why this?

Line 136-139 this method is not clear. Please state exactly what was done  

What volume of deionise was used and what was used to dry the glass ware? mention with details

Line 139-140 should be moved to line 136.

Line 145 to 146 ICP-MS technique poorly explained please improve

244 to 247 are not statistical analyses please delete

Line 253 to 254 what is it doing here? Delete

Lie 259 overhead 1.5°C ???? meaning?

Author Response

Dear reviewer,

Thank you for the attention with which you read the manuscript and for the recommendations and suggestions offered. We have made all the required changes, which significantly improved the quality of the manuscript, for which we express our gratitude. The references were double-checked in the main text and the Reference chapter. The manuscript was evaluated for the quality of English grammar by experts in the field; the certificate can be accessed on the platform. Point-by-point responses to comments are detailed in the following paragraphs.

This an interesting manuscript but the authors have over cited literature thus shadowing the relevance of the manuscript. The authors should go through the manuscript and remove many trivial sentences and revised the language. The figures should be improved. Below are some suggestions to help improved the manuscript.     

Response: Thank you very, much for accepting to evaluate our manuscript and offering suggestions and valuable recommendations which substantially improved our work. The figures were improved and an explanation was added to improve the clarity.

Line 17-18 the objective is not clear. Please rephrase to improve clarity

Response: We rephrased, “The warm trend of climate challenges the chemical risk associated with wine production around the globe. The present study investigated the possible difference between the chemical wine profile in the drought year 2012 compared to the post-drought year 2013.”

Line 19 A database that cumulated….meaning what? The abstract needs to be looked at. The current form is difficult to follow couple with grammar issues

Response: We rephrased.

The introduction is very long please cut some out. 25 citations in introduction alone is weird

Response: From the Introduction chapter was eliminated the sentences which discussed the general aspects related to wine and chemical profile. The statements were sustained with appropriate literature.

Line 114 to 123 please move elsewhere. Not appropriate after stating the aim of the study

Response: The paragraph was removed to improve the flow and clarity of the text. Thank you for the recommendation.

Line 127 assuming? Why this?

Response: The sentence was rephrased.

Line 136-139 this method is not clear. Please state exactly what was done  

Response: The method was detailed

What volume of deionise was used and what was used to dry the glass ware? mention with details

Response: We detailed in the text all methods to improve clarity and reproducibility.

Line 139-140 should be moved to line 136.

Response: The paragraph was moved to lime 136. Thank you!

Line 145 to 146 ICP-MS technique poorly explained please improve

Response:

244 to 247 are not statistical analyses please delete

Response: The paragraph was removed.

Line 253 to 254 what is it doing here? Delete

Response: The paragraph was removed.

Lie 259 overhead 1.5°C ???? meaning?

Response: The expression was changed. Thank you!

Reviewer 2 Report

Dear Authors and Editors

The article „Evaluating the chemical hazards in wine associated with climate change” was written in a transparent manner. In its assumptions, the study is interesting.

The topics presented in the article are appropriate for the Journal's profile.

The authors presented the discussed issues in a broad way.

The literature is selected in the right way.

The conclusions sum up the entire article appropriately.

However, I have a few technical notes:

- Figure 1,2,3,4,5,6 the quality of the graphs should be improved

- line 411 - is 3.5 ±08 should be 3.5 ± 0.8

In my opinion, this paper can be accepted for publication in Foods after minor corrects.

Best regards

Author Response

Dear Reviewer,

Thank you for the attention with which you read the manuscript and for the recommendations and suggestions offered. We have made all the required changes, which significantly improved the quality of the manuscript, for which we express our gratitude. The references were double-checked in the main text and the Reference chapter. In addition, the manuscript was evaluated for the quality of English grammar by experts in the field; the certificate can be accessed on the platform. Point-by-point responses to the comments are detailed below.

The authors presented the discussed issues in a broad way.

Response: Thank you very, much for accepting to evaluate our manuscript and offering suggestions and valuable recommendations which substantially improved our work.

The literature is selected in the right way.

Response: Thank you!

The conclusions sum up the entire article appropriately.

Response: Thank you!

However, I have a few technical notes:

- Figure 1,2,3,4,5,6 the quality of the graphs should be improved

Response: The figures were improved, and an explanation was added to improve the clarity. Thank you!

- line 411 - is 3.5 ±08 should be 3.5 ± 0.8

Response: Thank you for the attention to which you read the manuscript. 

Reviewer 3 Report

Foods-2253132

Evaluating the chemical hazards in wine associated with climate change

Material and Methods

When the analysis were performed?

How the wines were stored? The conditions were controlled and the same for all of them?

What are the conclusions?

Looks like there are no differences between the main factors studied.

Author Response

Dear Reviewer,

Thank you for the attention with which you read the manuscript and for the recommendations and suggestions offered. We have made all the required changes, which significantly improved the quality of the manuscript, for which we express our gratitude. The references were double-checked in the main text and the Reference chapter. In addition, the manuscript was evaluated for the quality of English grammar by experts in the field; the certificate can be accessed on the platform. Point-by-point responses to the comments are detailed below.

Material and Methods

When the analysis were performed?

Response: The analysis was performed in 2022. The information was added in the text.

How the wines were stored? The conditions were controlled and the same for all of them?

Response: Before preparation, the wine samples were briefly stored in a well-ventilated room with a controlled and constant temperature of approximately 14 °C. After the bottles were opened, the samples were immediately distributed to laboratories for multi-element and multi-isotope determinations. The answer was included in the main text of the manuscript. Thank you!

What are the conclusions?

Response: The following conclusions were added to the main text: “Various environmental factors strongly influence wine production, but climate change is responsible for various alterations, including increasing chemical hazards. The present survey investigated possible changes in the chemical composition of wine between the drought year (2012) and post-drought (2013). According to scPDSI indices, the Muntenia sites were most affected by drought in the early growing season of 2012, and the mean ground frost day count increased in 2013 in Transylvania. Minor changes in elemental profiles were found between years, and only K was found in an extremely high amount in 2012. Here, we report an extreme value of Cd (Stefanesti region) in Riesling wine, also noting that the Pb had a concentration above the values mentioned in the literature (‰). A significant difference between d18O in 2012 and 2013 was determined in Jidvei Sauvignon Blanc. On the basis of stable carbon (d13C of ethanol and glycerol) isotope ratios of wine samples, we found differences between Transylvania and Muntenia and also noted disentangled d13C values between years. The influence of environmental pollution on sites investigated was associated with yearly changes in 87Sr/86Sr and 206Pb/207Pb isotopic patterns. A strong significant correlation was found between climate variables and chemical profiles. Nevertheless, wine is a low-risk product, with the local enrichment of various chemicals, and no significant differentiation in terms of the elemental profile was achieved between years with clearly different climatic patterns.”

Looks like there are no differences between the main factors studied.

Response: Indeed, your conclusion is correct. The impact of the main environmental factors studied has no statistical significance on the chemical profile of the wine, regardless of the year of production.

Round 2

Reviewer 1 Report

The authors have modified the manuscript. Please see below minior comments. Accept the manuscript after these corrections are affected/ 

Line 22 and 23 remove the abbreviations

Line 290 A 1 ml aliquot  should be A aliquot (1 mL)

Line 291 to 292 should wine sample (1 mL) 69% nitric acid (0 mL)…………..Also what is the is the 69% ????

Line 312 A Sr isotope (25 μg/L)

Line 559 and 560 should be Vienna-Standard Mean Ocean Water (VSMOW) and Vienna-PeeDee Belemnite (VPDB)

Fugure 1 should be Mean climate differences between 2012 and 2013.

Line 137 delete of the  wine analyzed

Author Response

Dear reviewer,

We cordially express our gratitude for evaluating the manuscript a second time. We made all the changes according to your recommendations as follows.

The authors have modified the manuscript. Please see below minior comments. Accept the manuscript after these corrections are affected/ 

Line 22 and 23 remove the abbreviations

Response: The abbreviation was removed. Thank you!

Line 290 A 1 ml aliquot  should be A aliquot (1 mL)

Response: The sentence were changed at Line 125 and 209. Thank you!

Line 291 to 292 should wine sample (1 mL) 69% nitric acid (0 mL)…………..Also what is the is the 69% ????

Response: We rephrased as following ‘An aliquot (1 mL) of each sample was weighed, followed by digestion in a mixture of wine sample (1 mL) and nitric acid 69 % analytical purity (10 mL)’

Line 312 A Sr isotope (25 μg/L)

Response: We rephrased. Thank you!

Line 559 and 560 should be Vienna-Standard Mean Ocean Water (VSMOW) and Vienna-PeeDee Belemnite (VPDB)

Response: We rephrased.

Fugure 1 should be Mean climate differences between 2012 and 2013.

Response: In the figure 1,2,3 was illustrated the mean values and not median also the distribution statistics were added in the legend. The figure explanation was changed according to the recomenditions.

Line 137 delete of the wine analyzed

Response: The sentence was deleted. Thank you!